# The performance of HCV GT plus RUO reagent in determining Hepatitis C virus genotypes in Taiwan

Ying-Chou Huang[1], Chung-Feng Huang[1,2,3], Shu-Fen Liu[1], Hung-Yin Liu[1], Ming-Lun Yeh[1,2,3], Ching-I Huang[1,2,3], Meng-Hsuan Hsieh[1,2,3], Chia-Yen Dai[1,2,3], Shinn-Chern Chen[1,2,3], Ming-Lung Yu[1,2,3], Wan-Long Chuang[1,2,3], Jee-Fu Huang[1,2,3]*

**1** Hepatobiliary Division, Department of Internal Medicine, Kaohsiung Medical University Hospital, Kaohsiung, Taiwan, **2** Faculty of Internal Medicine, College of Medicine, Kaohsiung Medical University, Kaohsiung, Taiwan, **3** Hepatitis Centre, Kaohsiung Medical University Hospital, Kaohsiung, Taiwan

* jf71218@gmail.com

**Data Availability Statement:** All relevant data are within the paper.

**Funding:** The author(s) received no specific funding for this work.

## Abstract

### Background and aims

Hepatitis C virus (HCV) genotyping is a pivotal tool for epidemiological investigation, guiding management and antiviral treatment. Challenge existed in identifying subtypes of genotype-1 (G-1) and genotype (GT) of indeterminate. Recently, the Abbott HCV RealTime Genotype Plus RUO assay (HCV GT Plus) has been developed aiming to overcome the limitations. We aimed to evaluate the performance of the assay compared with 5' UTR sequencing in clinical samples.

### Materials and methods

Eligible individuals were treatment chronic hepatitis C patients that were enrolled consecutively in a medical center and two core regional hospitals in southern Taiwan from Oct 2017 through Aug 2018. The patient with genotype 1 without subtype and indeterminate previously genotyped by Abbott RealTime HCV GT II will further determinate by Abbott HCV RealTime HCV GT Plus. All of the genotype results were validated by 5' UTR sequencing as a reference standard.

### Results

A total of 100 viremic CHC patients were recruited, including 63 G-1 patients (male: 28), and 37 patients (male: 15) of indeterminate genotyped by Abbott RealTime HCV GT II assay (HCV GT II), respectively. The detection rate of 63 GT1 samples without subtype were 93.7% (59/63), 37 indeterminate samples without genotype were 62.2 (23/37) by HCV GT Plus. 5' UTR sequencing confirmed HCV GT Plus characterized results for 84.7% (50/59) of type1, with 100% (4/4), 82.8 (24/29) and 84.6% (22/26) for 1a, 1b and type6; 65.2% (15/23) of indeterminate with 100% (3/3) and 60% (12/20) for 1b and type 6 samples, respectively.

**Competing interests:** The HCV Genotype Plus RUO reagent provided by Abbott Molecular Diagnostics, Taiwan. This does not alter our adherence to PLOS ONE policies on sharing data and materials. The authors have declared that no competing interests exist.

## Conclusions

The Abbott RealTime HCV GT Plus RUO assay provides additional performance in GT detection.

## Introduction

Hepatitis C virus (HCV) is a 30–60 nm diameter, lipid-coated RNA virus and carries continuously a huge impact on liver-related events and hepatocellular carcinoma (HCC) globally [1]. According to the World Health Organization (WHO), there are 1.75 million new-infected cases annually, and still 75 million people have been suffering from chronic hepatitis C (CHC) worldwide. Approximately 399,000 people die directly due to CHC-related cirrhosis and HCC each year. The prevalence of anti-HCV seropositivity among adults in Taiwan was about 4.4%, much larger than other countries [2–4]. However, geographic difference of prevalence exists and it could reach to 15–20% in some hyperendemic areas in South Taiwan [5–7]. Therefore, research on HCV infection is of great significance to global health and public health.

Among all HCV genotypes, HCV genotype 1 (G-1) is the most prevalent in the world (46.2–49.1%), followed by G-3 (17.9–30.1%,), G-4 (8.3–16.8%), G-2 (9.1–11.0%), G-6 (5.4%), and G-5 (<1%) [8–10]. Apart from the implications of demographic epidemiology, different genotype may lead to different disease course and treatment outcomes. For example, G-1b leads to the highest incidence of developing cirrhosis and HCC [11–13]. The scenario becomes worse in heavy drinking, elders and HCV/ HIV co-infected [14–16]. G-3 infection is the next most prevalent genotype with 54.3 million patients globally. It is associated with an increasing risk of fibrosis, liver-related events, HCC, and overall mortality. G-3 infection may have a negative impact on histological and clinical outcomes in CHC patients. In addition, its characteristic features of steatosis and metabolic abnormalities may add more difficulty in the disease management [17–20]. Therefore, the precise determination of genotype in CHC patients is essential in a clinical setting. In addition, although G-1 and -2 were the major genotypes of HCV infection in Taiwan, geographic difference of genotype distribution occurs even in this beautiful country [21, 22].

In the past decade, direct-acting antivirals (DAAs), with the extremely high efficacy, low adverse effect, and short treatment duration, have become the main stream of HCV therapy. However, the successful treatment of HCV infection in some undeveloped and developing regions remains challenging because the cost and the affordability [23–25]. The precise diagnosis of genotyping is informative and critical for pretreatment assessment and subsequent therapeutic decision. To scale up HCV care cascade and HCV elimination, HCV genotyping may not be warranted at the population level. At the individual level, HCV genotyping might help to facilitate HCV treatment plan in terms of difficult-to-cure population (ex. HCV genotype 3) or to distinguish potential virological failure from reinfection. During the past decade, HCV genotyping technologies have been continuously improving, yielding many commercialized diagnostic assays for clinical purposes. Among them, Abbott RealTime HCV Genotype II Assay (HCV GT II, Abbott Laboratories, USA) is one of the widely-used assays for genotyping. However, clinical validation studies demonstrated that G-1a and G-1b subtypes could not be precisely identified among 6–7% G-1 (result showed type 1 only, no subtype-1a or 1b) samples, whereas 3–4% samples identified as indeterminate (detected HCV but did not produce a genotype result) genotype [26, 27].

In order to overcome the challenging issues, Abbott RealTime HCV Genotype Plus RUO assay (HCV GT Plus, Abbott Laboratories, USA), has been developed recently based on

Taqman real-time PCR method [28–30]. The performance verification of HCV GT Plus in Taiwan still has no complete evaluation report so far. Consequently, we conducted the study aiming to elucidate the performance of the HCV GT Plus in the genotyping diagnosis. Its detection performance will be compared with the direct sequencing results.

## Materials and methods

### Patient selection

This study was reviewed and approved by Institutional Review Board, Kaohsiung Medical University Hospital (KMUH/IRB/AF/2.8-01-11.0). The study was conducted in one medical centre and 2 regional hospitals in Southern Taiwan. We recruited those CHC patients with age and HCV viral loads criteria were over 20 years old above and over 500 IU/mL, respectively. All participants were informed of the research content and signed a written consent form. People with not detected for HCV RNA were suitable as negative control group. We also recruited the HCV genotype 1 with subtype 1a or 1b and genotype 6 previously genotyped by Abbott RealTime HCV GT II as positive control group. The patient's serum samples were determinate for genotype from Oct 2017 through Aug 2018. All the eligible patients with genotype 1 without subtype and indeterminate previously genotyped by Abbott RealTime HCV GT II will further determinate by Abbott HCV RealTime HCV GT Plus. All of the genotype results were validated by 5' UTR sequencing as a reference standard. The patients with other viral infections such as HIV or HBV were excluded.

### Sample preparation

All the serum samples were centrifuged within 4 hours after the blood was drawn, and the HCV viral load, genotype was tested within 1 week by HCV GT II by using Abbott RealTime HCV m2000 PCR System. The qualified samples were stored in -70˚C for further HCV GT Plus testing according to the manufacturer's manual. Direct sequencing was done on 5' UTR region by using ABI PRISM Big-Dye Terminator Cycle Sequencing Ready Reaction Kit, v3.1 (Applied Biosystems) on the ABI PRISM 3730XL DNA Analyzer. Primers used for PCR amplification and sequencing of 5' UTR region were TTGTGGTACTGCCTGATAGGG (forward) and GGATGTACCCCATGAGGTCG (reverse). The reverse transcription reaction was done by using High Capacity cDNA Reverse Transcription Kit according to the standard protocol of the supplier (Applied Biosystems) for 120 min at 37˚C. The clinical data such as albumin, bilirubin, transaminase, gamma-glutamyl transferase (GGT), and lipids levels were measured on a multi-channel auto analyzer (Hitachi Inc, Tokyo, Japan).

### Statistical analyses

Data were expressed as mean ± interquartile range (IQR). Chi-square test, Wicoxon rank-sum test, Pearson correlation coefficient and simple linear regression were used when appropriate. Quality control procedures, database processing and analyses were performed using the SPSS 20 statistical package (SPSS Inc., Chicago, IL, USA).

### Results

A total of 100 viremic CHC patients were recruited, including 63 G-1 patients, and 37 patients of indeterminate genotyped by Abbott RealTime HCV GT II assay (HCV GT II), respectively. Five healthy volunteers who were not infected with HCV served as a negative control group (NCG, HCV RNA: Not detected); for the positive control group, HCV GT Plus confirmed HCV GT II characterized results for 100% (3/3) of 1a, with 100% (2/2), 100% (3/3) for 1b and

type 6, respectively, validated by 5' UTR sequencing as a reference standard. Their demographical and clinical features of the patients are shown in Table 1.

For those 63 G-1 patients tested by HCV GT II, 59 (93.7%) patients were readily identified for their sub-genotypes or type by HCV GT Plus. There were 4 patients of G-1a, 29 patients of G-1b, 26 patients of G-6, and 4 patients of undetected, respectively. On the other side, for those 37 patients whose genotype showed indeterminate by HCV GT II, 23 (62.2%) patients could be genotyped by HCV GT Plus, including 3 patients of G-1b and 20 patients of G-6. We further used direct sequencing method based on 5' UTR sequence to test the performance of HCV GT Plus among those 63 G-1 patient samples detected by HCV GT II. The successful detection rate was 87.3% (55/63) by direct sequencing. The concordance rates between HCV GT Plus and direct sequencing method for G-1 samples was 84.8% (50/59), including 100% (4/4) of G-1a, 82.8% (24/29) of G-1b and 84.6% (22/26) of G-6.

Among those 26 patients who were genotyped G-1 by HCV GT II whereas G-6 genotype was identified on HCV GT Plus, 22 (84.6%) patients were of G-6 (G-6a = 7, 6e = 7, 6g = 4, 6u = 1, 6w = 3) by direct sequencing. Two of the 26 patients were genotyped as G-2 (Table 2).

There were 37 indeterminate genotype patients tested by HCV GT II. Twenty-three (62.2%) samples could be genotyped by HCV GT Plus, including 3 of G-1b, and 20 of G-6, respectively. Among those 20 patients who were genotyped indeterminate by HCV GT II whereas G-6 genotype was identified on HCV GT Plus, 12 (60%) patients were of G-6 (G-6a = 3, 6c = 4, 6e = 3, 6g = 1, 6n = 1) by direct sequencing. Among those 14 samples not detected by HCV GT Plus, 10 of them were genotyped as G-2 infection by direct sequencing (Table 3).

## Discussion

HCV strains are classified into eight recognized genotypes on the basis of phylogenetic and sequence analyses of whole viral genomes, further classified into 90 confirmed subtypes [31,

**Table 1. Basic demographic and clinical features of the two patient groups with type1 and indeterminate.**

|  | Type 1 (n = 63) | Indeterminate (n = 37) | p | NCG (n = 5) | PCG (n = 8) |
|---|---|---|---|---|---|
| Age (y), mean ± IQR | 60 ± 19 | 63 ± 15 | 0.62 | 55 ± 18 | 61 ± 25 |
| Sex, n (%) |  |  |  |  |  |
| Male | 28 (44%) | 15 (41%) |  | 2 (40%) | 2 (25%) |
| Female | 35 (56%) | 22 (59%) |  | 3 (60%) | 6 (75%) |
| WBC (×10$^3$/ μl) mean ± IQR | 5.33 ± 2.27 | 5.60 ± 2.57 | 0.39 | 5.33 ± 2.27 | 5.14 ± 0.64 |
| Hemoglobulin (g/dl) mean ± IQR | 12.3 ± 2.6 | 12.7 ± 1.6 | 0.70 | 13.2 ± 3.6 | 13.3 ± 3.1 |
| Platelet(×10$^3$/ μl) mean ± IQR | 152 ± 99 | 159 ± 101 | 0.71 | 169 ± 110 | 176 ± 100 |
| Albumin (g/dl) mean ± IQR | 4.1 ± 0.6 | 4.1 ± 0.7 | 0.63 | 4.0 ± 0.5 | 4.6 ± 0.3 |
| TBIL (mg/dl) mean ± IQR | 0.89 ± 0.57 | 1.00 ± 0.5 | 0.48 | 0.82 ± 0.6 | 0.80 ± 0.51 |
| DBIL (mg/dl) mean ± IQR | 0.15 ± 0.27 | 0.20 ± 0.30 | 0.77 | 0.18 ± 0.22 | 0.15 ± 0.06 |
| AST (IU/L) mean ± IQR | 41 ± 30 | 44 ± 49 | 0.64 | 39 ± 29 | 31 ± 20 |
| ALT (IU/L) mean ± IQR | 36 ± 40 | 46 ± 42 | 0.81 | 42 ± 41 | 27 ± 43 |
| GGT (IU/L) mean ± IQR | 24 ± 27 | 32 ± 47 | 0.08 | 36 ± 39 | 45 ± 42 |
| Creatinine (mg/dl) mean ± IQR | 0.8 ± 0.4 | 0.9 ± 0.4 | 0.36 | 1.0 ± 0.3 | 0.8 ± 0.2 |
| T-cholesterol (mg/dl) mean ± IQR | 167 ± 62 | 167 ± 43 | 0.34 | 192 ± 79 | 201 ± 69 |
| Triglyceride (mg/dl) mean ± IQR | 96 ± 46 | 167 ± 43 | 0.38 | 196 ± 82 | 123 ± 46 |
| HCV RNA (log**10** IU/mL) mean ± IQR | 5.77 ± 1.48 | 6.01 ± 0.72 | 0.18 | Not detected | 6.13 ± 0.70 |

$p > 0.05$, no significant difference (Wicoxon rank-sum test). IQR, interquartile range; WBC, white blood cell; TBIL, total bilirubin; DBIL, direct bilirubin; ALT, aspartate aminotransferase; AST, alanine aminotransferase; GGT, gamma-glutamyl transpeptidace; HCV, hepatitis C virus; NCG, negative control group; PCG, positive control group

**Table 2. Concordance comparison of HCV GT II/HCV GT *Plus* assays and direct sequencing results for the 63 G-1 patient samples.**

| HCV GT II | HCV GT *Plus* | | Direct sequencing | | Concordance rate between HCV GT *Plus* and direct sequencing (%) |
|---|---|---|---|---|---|
| G-1 (n = 63) | Genotype | n | Genotype | n | |
| | 1a | 4 | 1a | 4 | 100 |
| | 1b | 29 | 1b | 24 | 82.8 |
| | | | PCR failed | 5 | |
| | 6 | 26 | 6† | 22 | 84.6 |
| | | | 2 | 2 | |
| | | | PCR failed | 2 | |
| | Not detected | 4 | 1b | 3 | - |
| | | | PCR failed | 1 | |
| NCP (n = 5) | Not detected | 5 | Not detected | 5 | 100 |

†: 6a = 7, 6e = 7, 6g = 4, 6u = 1, 6w = 3

32]. HCV genotyping is essential in clinical setting for epidemiological study, implementation of therapeutic intervention, and outcome prediction [33–35]. Precise GT determination of genotypes by molecular diagnostic methods is a clinical challenging task with the current commercial assays, particularly in G-1 and indeterminate genotypes due to the high variability of HCV. The inability to distinguish G-6 subtypes might impact on redicted SVR to interferon (IFN)-based therapies in patients with apparent G-1 infection [35].

Taiwan government started to reimburse DAA with genotype specific regimens including daclatasvir /asunaprevir, paritaprevir/ritonavir/ombitasvir/dasabuvir, sofosbuvir/ledipasvir, sofosbuvir/ledipasvir and elbasvir/grazoprevir since 2017. For the pangenotypic DAAs, the National Health Insurance Administration of Taiwan start to reimburse glecaprevir/ pibrentasvir since Aug 2018 and sofosbuvir/velpatasvir since June 2019. Nowadays, available DAAs in Taiwan include elbasvir/grazoprevir, glecaprevir/ pibrentasvir and sofosbuvir/velpatasvir at the time of manuscript drafting. The choice of DAA regimens is at physicians' discretion. Even though pangenotypic DAAs are available in Taiwan, it is mandatory to provide and upload HCV genotype data to the government while applying DAAs per request by the authority (https://www.nhi.gov.tw/Content_List.aspx?n=A4EFF6CD1C4891CA&topn= 3FC7D09599D25979)

To our knowledge, this is the first study to address the characteristics of HCV GT Plus performance completely with G-1 and indeterminate patients in Taiwan. We demonstrated that

**Table 3. Concordance comparison of HCV GT II/HCV GT *Plus* assays and direct sequencing results for the 37 indeterminate genotype patient samples.**

| HCV GT II | HCV GT *Plus* | | Direct sequencing | | Concordance rate between HCV GT *Plus* and direct sequencing (%) |
|---|---|---|---|---|---|
| Indeterminate (n = 37) | Genotype | n | Genotype | n | |
| | 1a | 0 | 1a | 0 | - |
| | 1b | 3 | 1b | 3 | 100 |
| | 6 | 20 | 6‡ | 12 | 60 |
| | | | PCR failed | 8 | |
| | Not detected | 14 | 2 | 10 | - |
| | | | PCR failed | 4 | |
| NCP (n = 5) | Not detected | 5 | Not detected | 5 | 100 |

‡: 6a = 3, 6c = 4, 6e = 3, 6g = 1, 6n = 1

the concordance rate was only 52.4% (33/63) between HCV GT II and HCV GT Plus in G-1 patients. G-6 and its variants, previously mistyped as G-1. HCV G-6, highly prevalent in the Asia–Pacific region, can be incorrectly classified as G-1 because they share similar nucleotide homology in the 5′UTR [36, 37]. Mainly infection areas of G-6 are concentrated in South Asia, including Vietnam, Thailand, Hong Kong and Macau [38, 39]. Ethnic groups are mainly prevalent in injection drug users (IDUs). The main infection subtype of G-6 in Taiwan was 6a [40], but this time we also found other G-6 subtypes (6c, 6e, 6g, 6n).

Direct sequencing confirmed HCV GT Plus results for 84.7% (50/59) of G-1, with 100% (4/4), 82.8% (24/29) and 84.6% (22/26) for 1a, 1b and GT 6, respectively. Therefore, the HCV GT Plus assay has a very good ability to distinguish 1a, 1b and G-6 in G-1 group.

Among the 37 patients of indeterminate genotype detected by HCV GT II, 62.2% (23/37) patients were readily identified for their sub-genotypes or type by HCV GT Plus. The concordance rate increased 65.2% (15/23) between HCV GT Plus and direct sequencing method. Our results thus provided the comparison between the 2 assays in clinical application. The discordant results between assays may also raise the concern of potential limitations for clinical precise diagnosis.

The Abbott Realtime HCV Plus RUO assay is an automated HCV genotyping method specifically targets the HCV core region with multiple TaqMan probes for HCV genotypes 1a, 1b, and 6. Addition of Core motifs could improve the discrimination between G-1 and G-6 [41].

Our results found that HCV GT Plus for G-1 group whether in the detection rate (93.7% vs. 62.2%, $p = 0.10$) or concordance (84.7% vs. 65.2%, $p = 0.009$) was better than indeterminate group.

The samples that have preliminary results from HCV GT II, the higher proportions genotype results will obtain by HCV Plus or direct sequencing.

For the indeterminate group, the detection rate was only 62.2%, 10 of 14 samples (37.8%) showed not detected due to 15 subtypes of type 2 [37] that HCV GT II cannot cover all subtypes and HCV GT Plus enhancements for 1a, 1b and type 6 only.

The HCV GT Plus could also potentially be used to resolve high rate (up to 34.9%) of misidentifying genotype 6 samples as genotype 1, confirmed by 5′ UTR direct sequencing.

A variety of genotyping auxiliary methods (such as Okamoto typing or direct sequencing) have limitations in the past. Okamoto typing can only detect 1a, 1b, 2a, 2b and 3a, respectively [21]. Direct sequencing will also have some differences in the results due to the region selected by the primer's position (5′untranslated region, core or Nonstructural protein 5B [42, 43].

Despite design limitations (core region for 1a, 1b, and 6 only, but 99% to be attributable to subtypes 1a and 1b and inability to detect all subtype of G-1 and G-6, the HCV Plus completes HCV GT II well and thus helps to overcome the shortcomings of the HCVGT II assay design, significantly reduced the frequency of G-1 without subtype results, while also providing the ability to identify G-6, reliably characterized confirmed by 5′ UTR sequencing. Regardless of cost or time spent, HCV GT Plus assay is lower than direct sequencing. As far as auxiliary testing is concerned, HCV GT Plus assay is more convenient and cheaper than direct sequencing. Therefore, we recommend that Abbott RealTime HCV Genotype II Assay should be collocated with HCV GT Plus assay.

## Conclusions

The shortcomings of Abbott RealTime HCV GT II assay without subtype (ST) of type1 and genotype (GT) of indeterminate can be further resolved and reliably characterized by the new Abbott RealTime HCV GT Plus RUO assay in certain ambiguous samples.

## Acknowledgments

We are very grateful to Pey-Fang Wu and Yu-Fen Wang for giving suggestions to improve the manuscript.

## Author Contributions

**Conceptualization:** Ying-Chou Huang, Chung-Feng Huang.

**Data curation:** Ying-Chou Huang, Chung-Feng Huang, Jee-Fu Huang.

**Formal analysis:** Ying-Chou Huang, Chung-Feng Huang.

**Investigation:** Ching-I Huang.

**Methodology:** Shu-Fen Liu, Ching-I Huang.

**Project administration:** Shu-Fen Liu.

**Resources:** Hung-Yin Liu, Ming-Lun Yeh, Meng-Hsuan Hsieh, Jee-Fu Huang.

**Software:** Hung-Yin Liu, Ming-Lun Yeh, Meng-Hsuan Hsieh, Shinn-Chern Chen.

**Supervision:** Chia-Yen Dai, Ming-Lung Yu, Wan-Long Chuang, Jee-Fu Huang.

**Validation:** Ying-Chou Huang, Chia-Yen Dai, Shinn-Chern Chen, Ming-Lung Yu, Wan-Long Chuang, Jee-Fu Huang.

**Visualization:** Ming-Lun Yeh.

**Writing – original draft:** Ying-Chou Huang.

**Writing – review & editing:** Jee-Fu Huang.

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
