## [Decision Letter · Decision Letter 0]

21 Dec 2020

PONE-D-20-35973

The performance of HCV GT Plus RUO reagent in determining hepatitis C virus genotypes in Taiwan

PLOS ONE

Dear Dr. Huang,

Thank you for submitting your manuscript to PLOS ONE. After careful consideration, we feel that it has merit but does not fully meet PLOS ONE’s publication criteria as it currently stands. Therefore, we invite you to submit a revised version of the manuscript that addresses the points raised during the review process both in the methodology section and also editing of the manuscript.

We look forward to receiving your revised manuscript.

Kind regards,

Isabelle Chemin, PhD

Academic Editor

PLOS ONE

Journal Requirements:

"We gratefully thank the secretary help from Taiwan Liver Research Foundation

(TLRF) and the reagents provided by Abbott Molecular Diagnostics, Taiwan. They

did not influence how the study was conducted or the approval of the manuscript."

Additionally, because some of your funding information pertains to commercial funding, we ask you to provide an updated Competing Interests statement, declaring all sources of commercial funding.

In your Competing Interests statement, please confirm that your commercial funding does not alter your adherence to PLOS ONE Editorial policies and criteria by including the following statement: "This does not alter our adherence to PLOS ONE policies on sharing data and materials.” as detailed online in our guide for authors  http://journals.plos.org/plosone/s/competing-interests.  If this statement is not true and your adherence to PLOS policies on sharing data and materials is altered, please explain how.

Please include the updated Competing Interests Statement and Funding Statement in your cover letter. We will change the online submission form on your behalf.

4. Please amend the manuscript submission data (via Edit Submission) to include author Zu-Yau Lin.

Reviewers' comments:

Reviewer's Responses to Questions

**Comments to the Author**

1. Is the manuscript technically sound, and do the data support the conclusions?

Reviewer #1: Yes

2. Has the statistical analysis been performed appropriately and rigorously? 

Reviewer #1: N/A

3. Have the authors made all data underlying the findings in their manuscript fully available?

Reviewer #1: Yes

4. Is the manuscript presented in an intelligible fashion and written in standard English?

Reviewer #1: Yes

5. Review Comments to the Author

Reviewer #1: This is a relatively straightforward study comparing two Abbott genotyping assays. The authors found that the new assay was able to correctly “genotyped” some of those that was either indeterminate or genotype 1 but no subtype using the new Plus assays, using sequencing as the gold standard.

1. There should be another control group, comparing the results between the Abbott Real Time HCV GT II with the Plus assays among those that were not indeterminate or no subtype. The study found that among those that were genotype (no subtype), significant numbers turned out to be genotype 6. Among those that were genotyped as 1a, 1b, 1c etc by the Abbott Real Time HCV GT II or Plus, what percentage turned out to be genotype 6? Does this a more common findings for those indeterminate sample only? This is an important question since the authors emphasized the importance of genotyping correctly.

2. For those no subtype or indeterminate, were they repeated again with GT II with same result or tested only once?

3. Among the G1 no subtype and in the indeterminate group, 8 samples in each group PCR failed. Was it due to the condition of the sample? This is one of the limitations of the study. This is particularly problematic for the indeterminate group, since this represented almost one quarter of the sample.

4. In the G1 no subtype group, 4 were “undetected” by the Plus assay. Were these undetected or indeterminate by the Plus assay?

5. The discussion should include existing literature on mistyping/problem with correctly identifying genotype 6. Currently, this was mentioned only briefly with only one reference (#35). This is an important point since genotype 6 is common in South Asia.

6. In the introduction section, the authors stated that “The precise diagnosis of genotyping is informative and critical for pretreatment assessment and subsequent therapeutic decision”. This sentence is actually incorrect. The significance of current study was markedly reduced with pangenotypic DAA regimen. EASL guidelines actually suggest genotyping is optimal in resources limited countries and should not be a barrier to DAA treatment, even though genotyping might be important. The MINMON studies reported an SVR rate of 95% with SOF/VEL in over 300 subjects without genotyping.

7. It would help the readers to understand better the significance of this study if the authors could also briefly describe the treatment landscape in Taiwan and insurance reimbursement scheme. Was the choice of DAA depend on the genotype instead of pangenotypic regimen as first line therapy? Was genotype required for reimbursement? Was there major cause differences between the pangenotypic DAA vs earlier DAA that were genotype specific?

6. PLOS authors have the option to publish the peer review history of their article (what does this mean?). If published, this will include your full peer review and any attached files.

Reviewer #1: No

---

## [Author Response · Author response to Decision Letter 0]

12 Jan 2021

PONE-D-20-35973

The performance of HCV GT Plus RUO reagent in determining hepatitis C virus genotypes in Taiwan

Review Comments to the Author

Reviewer #1: This is a relatively straightforward study comparing two Abbott genotyping assays. The authors found that the new assay was able to correctly “genotyped” some of those that was either indeterminate or genotype 1 but no subtype using the new Plus assays, using sequencing as the gold standard.

1. There should be another control group, comparing the results between the Abbott Real Time HCV GT II with the Plus assays among those that were not indeterminate or no subtype. The study found that among those that were genotype (no subtype), significant numbers turned out to be genotype 6. Among those that were genotyped as 1a, 1b, 1c etc by the Abbott Real Time HCV GT II or Plus, what percentage turned out to be genotype 6? Does this a more common findings for those indeterminate sample only? This is an important question since the authors emphasized the importance of genotyping correctly.

Reply: Thank you for the insightful comment. We fully agree with the reviewer that there should be another control group, comparing the results between the Abbott Real Time HCV GT II with the Plus assays among those that were not indeterminate or no subtype. We would like to add “We also recruited the HCV genotype 1 with subtype 1a or 1b and genotype 6 previously genotyped by Abbott RealTime HCV GT II as the positive control group.” in the Materials and Methods section and “For the positive control group, HCV GT Plus confirmed HCV GT II characterized results for 100% (3/3) of 1a, with 100% (2/2), 100% (3/3) for 1b and type 6, respectively, validated by 5' UTR sequencing as a reference standard” in the Results section.” 

As subtype 1a and 1b previously genotyped by Abbott RealTime HCV GT II, would not be considered as genotype 6 by HCV GT Plus or direct sequencing. The misidentifying issue only occurred in genotype 1 without subtype 1a or 1b previously genotyped by Abbott RealTime HCV GT II. According to the data from our research , 54.1% (20/37) were tested as genotype 6 in indeterminate group, higher than 41.3% (26/63) misidentifying genotype 6 samples as genotype 1 without subtype 1a or 1b, genotyped by Abbott RealTime HCV GT Plus.

2. For those no subtype or indeterminate, were they repeated again with GT II with same result or tested only once?

Reply: Thank you for the comment. All samples were tested only once by GT II.

3. Among the G1 no subtype and in the indeterminate group, 8 samples in each group PCR failed. Was it due to the condition of the sample? This is one of the limitations of the study. This is particularly problematic for the indeterminate group, since this represented almost one quarter of the sample.

Reply: Thank you for the insightful comment. We fully agree with the reviewer that PCR failed will be the limitations of the study. All samples have been checked for quality before processing. The condition of all samples were normal without hemolysis, lipemia or jaundice in our study. The qualified samples were stored in -70 ° C for further HCV GT Plus testing according to the manufacturer’s manual. 

4. In the G1 no subtype group, 4 were “undetected” by the Plus assay. Were these undetected or indeterminate by the Plus assay?

Reply: Thank you for the comment. These four samples were tested by the Plus assay, and the results showed not detected. 

5. The discussion should include existing literature on mistyping/problem with correctly identifying genotype 6. Currently, this was mentioned only briefly with only one reference (#35). This is an important point since genotype 6 is common in South Asia.

Reply: Thank you for the insightful comment. We retrieve the existing literature on mistyping/problem with correctly identifying genotype 6 from PubMed, which is organized by National Center for Biotechnology Information (NCBI) of the United States of America. We would like to add the description “ The inability to distinguish HCV-6 subtypes might impact on predicted SVR to interferon (IFN)-based therapies in patients with apparent HCV-1 infection[1]. HCV G-6, highly prevalent in the Asia–Pacific region, can be incorrectly classified as HCV-1b because they share similar nucleotide homology in the 5′UTR [2]. Addition of Core motifs could improve the discrimination between HCV-1 and HCV-6 [3].” in Discussion section 

6. In the introduction section, the authors stated that “The precise diagnosis of genotyping is informative and critical for pretreatment assessment and subsequent therapeutic decision”. This sentence is actually incorrect. The significance of current study was markedly reduced with pangenotypic DAA regimen. EASL guidelines actually suggest genotyping is optimal in resources limited countries and should not be a barrier to DAA treatment, even though genotyping might be important. The MINMON studies reported an SVR rate of 95% with SOF/VEL in over 300 subjects without genotyping.

Reply: Thank you for the insightful comment. We fully agree with the reviewer that HCV genotyping might not be warranted in the era with pangenotypic DAA regimens. We have omitted the description. We would like to add the description “ To scale up HCV care cascade and HCV elimination, HCV genotyping may not be warranted at the population level. At the individual level, HCV genotyping might help to facilitate HCV treatment plan in terms of difficult-to-cure population (ex. HCV genotype 3) or to distinguish potential virological failure from reinfection.” 

7. It would help the readers to understand better the significance of this study if the authors could also briefly describe the treatment landscape in Taiwan and insurance reimbursement scheme. Was the choice of DAA depend on the genotype instead of pangenotypic regimen as first line therapy? Was genotype required for reimbursement? Was there major cause differences between the pangenotypic DAA vs earlier DAA that were genotype specific?

Reply: Thank you for the insightful comment. Taiwan government started to reimburse DAA with genotype specific regimens including daclatasvir /asunaprevir, paritaprevir/ritonavir/ombitasvir/dasabuvir, sofosbuvir/ledipasvir, sofosbuvir/ledipasvir and elbasvir/grazoprevir since 2017. For the pangenotypic DAAs, the National Health Insurance Administration of Taiwan start to reimburse glecaprevir/ pibrentasvir since Aug 2018 and sofosbuvir/velpatasvir since June 2019. Nowadays, available DAAs in Taiwan include elbasvir/grazoprevir, glecaprevir/ pibrentasvir and sofosbuvir/velpatasvir at the time of manuscript drafting. The choice of DAA regimens is at physicians’ discretion. Even though pangenotypic DAAs are available in Taiwan, it is mandatory to provide and upload HCV genotype data to the government while applying DAAs per request by the authority (https://www.nhi.gov.tw/Content_List.aspx?n=A4EFF6CD1C4891CA&topn=3FC7D09599D25979). 

References

1. Dev AT, McCaw R, Sundararajan V, Bowden S, Sievert W. Southeast Asian patients with chronic hepatitis C: the impact of novel genotypes and race on treatment outcome. Hepatology (Baltimore, Md). 2002;36(5):1259-65. 

2. McCaughan GW, Omata M, Amarapurkar D, Bowden S, Chow WC, Chutaputti A, et al. Asian Pacific Association for the Study of the Liver consensus statements on the diagnosis, management and treatment of hepatitis C virus infection. Journal of gastroenterology and hepatology. 2007;22(5):615-33. 

3. Bouchardeau F, Cantaloube JF, Chevaliez S, Portal C, Razer A, Lefrère JJ, et al. Improvement of hepatitis C virus (HCV) genotype determination with the new version of the INNO-LiPA HCV assay. Journal of clinical microbiology. 2007;45(4):1140-5.

---

## [Decision Letter · Decision Letter 1]

18 Jan 2021

The Performance of HCV GT Plus RUO Reagent in Determining Hepatitis C Virus Genotypes in Taiwan

PONE-D-20-35973R1

Dear Dr. Huang,

We’re pleased to inform you that your manuscript has been judged scientifically suitable for publication and will be formally accepted for publication once it meets all outstanding technical requirements.

Kind regards,

Isabelle Chemin, PhD

Academic Editor

PLOS ONE

Additional Editor Comments (optional):

Reviewers' comments:

Reviewer's Responses to Questions

**Comments to the Author**

1. If the authors have adequately addressed your comments raised in a previous round of review and you feel that this manuscript is now acceptable for publication, you may indicate that here to bypass the “Comments to the Author” section, enter your conflict of interest statement in the “Confidential to Editor” section, and submit your "Accept" recommendation.

Reviewer #1: All comments have been addressed

2. Is the manuscript technically sound, and do the data support the conclusions?

Reviewer #1: Yes

3. Has the statistical analysis been performed appropriately and rigorously? 

Reviewer #1: N/A

4. Have the authors made all data underlying the findings in their manuscript fully available?

Reviewer #1: Yes

5. Is the manuscript presented in an intelligible fashion and written in standard English?

Reviewer #1: Yes

6. Review Comments to the Author

Reviewer #1: No new comments to the authors.

7. PLOS authors have the option to publish the peer review history of their article (what does this mean?). If published, this will include your full peer review and any attached files.

Reviewer #1: No

---

## [Editor Report · Acceptance letter]

21 Jan 2021

PONE-D-20-35973R1 

The Performance of HCV GT Plus RUO Reagent in Determining Hepatitis C Virus Genotypes in Taiwan 

Dear Dr. Huang:

I'm pleased to inform you that your manuscript has been deemed suitable for publication in PLOS ONE. Congratulations! Your manuscript is now with our production department. 

Kind regards, 

on behalf of

Mrs Isabelle Chemin 

Academic Editor

PLOS ONE